# Mucoadhesive, Thermoreversible Hydrogel, Containing Tetracaine-Loaded Nanostructured Lipid Carriers for Topical, Intranasal Needle-Free Anesthesia

**DOI:** 10.3390/pharmaceutics13111760

**Published:** 2021-10-21

**Authors:** Giovana Maria Fioramonti Calixto, Bruno Vilela Muniz, Simone R. Castro, Jaiza Samara Macena de Araujo, Klinger de Souza Amorim, Lígia N. M. Ribeiro, Luiz Eduardo Nunes Ferreira, Daniele Ribeiro de Araújo, Eneida de Paula, Michelle Franz-Montan

**Affiliations:** 1Department of Biosciences, Piracicaba Dental School, University of Campinas-UNICAMP, Piracicaba 13414-903, Brazil; giovana.calixto@gmail.com (G.M.F.C.); bruno.vilela@professor.fait.edu.br (B.V.M.); j181581@dac.unicamp.br (J.S.M.d.A.); k193047@dac.unicamp.br (K.d.S.A.); 2Department of Biochemistry and Tissue Biology, Institute of Biology, University of Campinas-UNICAMP, Campinas 13083-872, Brazil; simonercastro@hotmail.com (S.R.C.); depaula@unicamp.br (E.d.P.); 3Institute of Biotechnology, Federal University of Uberlandia-UFU, Uberlandia 38405-302, Brazil; ligia.ribeiro@ufu.br; 4Laboratory of Inflammation and Immunology, Guarulhos University-UNG, Guarulhos 07023-070, Brazil; luiz.ferreira@prof.ung.br; 5Human and Natural Sciences Center, Federal University of ABC-UFABC, Santo André 09210-580, Brazil; daniele.araujo@ufabc.edu.br

**Keywords:** nanostructured lipid carriers, thermoreversible hydrogel, nasal administration, mucoadhesion, topical anesthesia, tetracaine

## Abstract

Recent advances have been reported for needle-free local anesthesia in maxillary teeth by administering a nasal spray of tetracaine (TTC) and oxymetazoline, without causing pain, fear, and stress. This work aimed to assess whether a TTC-loaded hybrid system could reduce cytotoxicity, promote sustained permeation, and increase the anesthetic efficacy of TTC for safe, effective, painless, and prolonged analgesia of the maxillary teeth in dental procedures. The hybrid system based on TTC (4%) encapsulated in nanostructured lipid carriers (NLC) and incorporated into a thermoreversible hydrogel of poloxamer 407 (TTC_NLC-HG4%_) displayed desirable rheological, mechanical, and mucoadhesive properties for topical application in the nasal cavity. Compared to control formulations, the use of TTC_NLC-HG4%_ slowed in vitro permeation of the anesthetic across the nasal mucosa, maintained cytotoxicity against neuroblastoma cells, and provided a three-fold increase in analgesia duration, as observed using the tail-flick test in mice. The results obtained here open up perspectives for future clinical evaluation of the thermoreversible hybrid hydrogel, which contains TTC-loaded NLC, with the aim of creating an effective, topical, intranasal, needle-free anesthesia for use in dentistry.

## 1. Introduction

Needle-free dental anesthesia is a promising alternative to conventional local anesthesia, especially for needle-phobic patients, because it can overcome issues associated with the anxiety and fear of needles [1,2]. In dentistry, a significant advance in needle-free anesthesia was achieved with the development of a nasal spray containing 3% tetracaine (TTC) plus 0.05% oxymetazoline, called Kovanaze^®^ (St. Renatus, LLC, Fort Collins, CO, USA), for local anesthesia of maxillary anterior and first premolar teeth [3].

Topical intranasal administration is an attractive approach for anesthetic delivery, since local anesthetics (LA) can diffuse through the thin nasal mucosa, blocking nociceptive impulses from the maxillary teeth and surrounding tissues, consequently generating effective anesthesia for performing dental restorative procedures in maxillary premolars, canines, and incisors [2,3,4]. Furthermore, the increased permeability of the nasal mucosa can facilitate the fast onset of action of the LA [5,6].

Despite these advantages, a single-blind crossover study, which investigated the efficacy of Kovanaze^®^ nasal spray for maxillary teeth pulpal anesthesia, showed that after nasal topical administration, the volunteers experienced nasal drainage, nasal congestion, nasal burning, and sinus pressure and congestion. Moreover, after experiencing both routes of administration (topical nasal spray and infiltration injection with 2% lidocaine plus 1:100,000 epinephrine), 100% of the volunteers indicated preference for the traditional injection [1]. Other studies have reported additional side effects, such as rhinorrhea or nasal discomfort [3,4].

Another issue that needs to be considered in order to achieve effective pulpal and soft tissue anesthesia is the ability of the formulation to stay adhered at the application site, since physical removal of the drug by the mucociliary clearance mechanisms, together with enzymatic degradation in the mucus layer, may limit intranasal permeation [5,6,7]. One strategy that can improve nasal absorption and provide a longer duration of analgesia is to encapsulate the LA into nanocarriers, such as nanostructured lipid carriers (NLC) [8,9]. NLC are colloidal delivery systems with high permeability efficiency and safety, scale-up feasibility, biocompatibility, biodegradability, and low toxicity [10].

With these issues in mind, our research group developed a TTC-in-NLC formulation, optimized by means of a 2^3^ factorial design [11]. This optimized formulation, which contains TTC at the clinical dose of 4%, showed excellent structural properties and high TTC loading capacity. Furthermore, the NLC provided sustained release beyond 48 h, reduced cytotoxicity against cultured Balb/c 3T3 fibroblasts, and stability for up to 365 days at 25 °C [11].

To make this TTC-in-NLC formulation suitable for topical drug delivery, we dispersed it in a mucoadhesive and thermosensitive hydrogel which was produced using poloxamer 407 (PL407). PL407 is a non-ionic, water-soluble triblock copolymer with hydrophobic polyoxypropylene (PPO) chains in between two hydrophilic polyoxyethylene (POE) blocks. Its thermoresponsive behavior is due to its capacity to self-assemble into micelles in aqueous solutions at specific concentrations and temperatures [12,13]. The critical micellar concentration at which to start the PL407 micellization process decreases with an increase in temperature due to the difference in the solvation of the POE and PPO blocks. This favors hydrophobic interactions among the PPO blocks and the formation of spherical micelles, followed by a transition from the liquid to the gel phase, at physiological temperatures [14]. Moreover, PL407 hydrogels can form entanglements or non-covalent bonds with mucus, which decrease mucociliary clearance and prolong the residence time at the nasal mucosa [15,16].

The aim of the present work was to assess whether a hybrid system based on TTC-loaded NLC, further incorporated into PL407 hydrogel, could reduce the cytotoxicity of tetracaine, while increasing its permeation capacity and anesthetic efficacy.

## 2. Materials and Methods

### 2.1. Materials

Tetracaine hydrochloride was purchased from AK Scientific, Inc. (Union City, CA, USA); Dhaykol 6040 LW was purchased from Dhaymers Química Fina (Taboão da Serra, Brazil); DMEM/F12-Ham (Dulbecco’s Modified Eagle Medium) was from Nutricell (Campinas, Brazil); myristyl myristate, poloxamer 188 (PL188), and poloxamer 407 (PL407) were purchased from Sigma Chem. Co. (St. Louis, MO, USA).

### 2.2. Analytical Procedures

Tetracaine quantification during the encapsulation efficiency and in vitro permeation studies was performed using a high performance liquid chromatography (HPLC) system (Varian ProStar, equipped with a PS 325 UV-Vis detector, PS 210 solvent delivery module with Galaxie™ Chromatography Workstation, Varian, Inc. Walnut Creek, CA, USA). Briefly, a Luna (5μm, 250 × 4.6 mm) reversed-phase C18 column (Phenomenex^®^, Torrance, CA, USA) was employed, with an isocratic elution employing a mobile phase composed of ammonium phosphate (10 mM, pH 3.0): acetonitrile, 70:30 (*v*/*v*). The detection wavelength, eluent flow rate, and injection volume were set at 280 nm, 1.5 mL/min, and 30 µL, respectively. The method was successfully validated, with a linear calibration curve (r^2^ = 0.9998), in the concentration range from 0.03 mM to 1.2 mM. The TTC retention time was 5.3 ± 0.1 min and the TTC limits of quantification and detection were 1.5 × 10^−4^ mM 4.5 × 10^−5^ mM, respectively.

### 2.3. Preparation of the Formulation

TTC-loaded NLC (TTC_NLC_) were prepared using the ultrasound emulsification method [17]. Briefly, the oil phase was prepared by mixing the myristyl myristate (1.22 g) and Dhaykol 6040 LW (0.48 g) lipids at 50 °C under continuous agitation to achieve a homogeneous mixture. Then, TTC base (4%, 0.4 g) was added to the melted lipid phase and kept under stirring until complete dissolution. Separately, an aqueous phase was prepared by dissolving the PL188 surfactant (0.5 g for each 10 mL of deionized water), at 50 °C, followed by its addition to the lipid phase and homogenization using an Ultra-Turrax (IKA T18 basic, WerkeStaufen, Germany) at 10,000 rpm, for 3 min. The O/W pre-emulsion obtained was kept under heating and was tip sonicated (Vibracell, Sonics & Materials Inc., Danbury, CT, USA) at 50 W and 20 kHz, for 30 min, with 30 on/off cycles of 30 s. The O/W nanoemulsion formed was immediately cooled to 25 °C in an ice bath, giving rise to the nanoparticles (TTC_NLC4%_). A control formulation without TTC (NLC) was prepared, as well as (only for the tail-flick tests) TTC_NLC2%_ and TTC 2% samples.

The particle size and polydispersity index (PDI) of the NLC formulation were determined by dynamic light scattering (DLS). The zeta potential (ZP) was calculated using the electrophoretic mobility method, using a Nano ZS90 analyzer (Malvern Instruments, UK), at 25 °C [11]. The samples were diluted (1000× in deionized water) and the measurements were performed in triplicate.

The TTC encapsulation efficiency (%EE) was measured by the ultrafiltration-centrifugation method [10], in triplicate, using regenerated cellulose filters (Millipore^®^, Darmstadt, Germany) with a molecular exclusion pore size of 10 kDa. The diluted samples were centrifuged at 6000× *g* for 20 min. The filtered, aqueous, non-encapsulated fractions of TTC were quantified by HPLC. The %EE index of TTC (%) was determined according to Equation (1):(1)$$\%\mathrm{EE}=\frac{{\mathrm{TTC}}_{\mathrm{total}}-{\mathrm{TTC}}_{\mathrm{free}\;}}{{\mathrm{TTC}}_{\mathrm{total}}}\times 100$$
where:

TTC_total_ = Total amount of TTC 

TTC_free_ = Non-encapsulated fraction of TTC

To prepare the hydrogels containing TTC_NLC2%_ and TTC_NLC4%_ (TTC_NLC-HG2%_ and TTC_NLC-HG4%_), 25% poloxamer PL407 was first dispersed in deionized water, at 4 °C, under mechanical stirring (300 rpm) for 2 h [18,19,20]. A hydrogel without TTC (HG) was also prepared for tail flick tests. To obtain the hybrid hydrogel with 4% TTC (TTC_NLC-HG4%_), 4% tetracaine hydrochloride-loaded hydrogel (TTC_HG4%_) was first prepared, followed by mixing 25 mL of TTC_HG4%_ with 25 mL of TTC_NLC4%_, resulting in TTC_NLC-HG4%_. To obtain the hybrid hydrogel with 2% TTC (TTC_NLC-HG2%_), used in the antinociceptive tests, 25 mL of TTC_NLC4%_ was added to 25 mL of PL407 hydrogel, resulting in TTC_NLC-HG2%_. Hydrogel loaded with 2% tetracaine hydrochloride (TTC_HG2%_) was also prepared. Finally, hybrid hydrogel without TTC loading (HG+NLC) was prepared by mixing 25 mL of NLC (without TTC) and 25 mL of PL407 hydrogel. The TTC_NLC_:TTC_HG_ ratio in the final hybrid hydrogel loaded with 4% TTC was 1:1 (*v*:*v*).

### 2.4. Characterization of the Formulations

#### 2.4.1. Rheological Analysis

Rheological analyses were performed using an oscillatory rheometer (model Kinexus, Malvern Instruments Ltd., Malvern, UK) with cone-plate geometry (model CP4/40 SR2577SS, plate size of 40 mm, sample volume of 1 mL, gap between plates of 1 mm, and a 4° cone angle). For determination of the sol-gel transition temperature (T_sol-gel_), the frequency (f) was set to 1 Hz and a temperature interval from 10 to 50 °C was selected. Temperature sweep analysis was performed for 30 min, with variation of the temperature at a rate of 5 °C/min. The sol-gel transition temperature was determined as the point where more pronounced viscosity variation was observed during the temperature sweep. In addition, for frequency sweep analysis, the formulations (TTC_HG4%_ and TTC_NLC-HG4%_) were analyzed from 0.1 to 10 Hz, at 32.5 °C [20]. All the analyses were performed at 32.5 °C, which was considered to be the mean nasal mucosa temperature [21].

#### 2.4.2. Texture Profile Analysis (TPA)

TPA was performed using a TA-XT Plus texture analyzer (Stable Micro Systems, Surrey, Godalming, UK) operated in TPA mode. Samples (10 g) of TTC_HG4%_ and TTC_NLC-HG4%_ were carefully transferred to vials and maintained in a water bath at 37 °C for 24 h. After that period, the cylindrical analytical probe (10 mm diameter) was lowered (at 2 mm·s^−1^) until it reached the sample. The formulations were compressed twice (0.5 mm·s^−1^, 5 mm depth, 15 s delay period). Cohesiveness (the work during the second compression divided by the work during the first compression), adhesiveness (the force per unit time required to detach the probe from the formulation during the first compression), and elasticity (the time interval during the second compression divided by the time interval during the first compression) were calculated from the force-time curves, using Expert Texture Exponent 32 software (version 6.1.18.0; Stable Micro Systems, Surrey, UK) [22].

#### 2.4.3. In Vitro Mucoadhesive and Permeation Studies

##### Preparation of Porcine Nasal Mucosa

The nasal mucosa used in the mucoadhesion and permeation experiments were obtained from five-month-old pigs weighing between 70–80 kg. The mucosa were supplied by the Angelelli^®^ slaughterhouse (certified by the Brazilian Ministry of Agriculture) shortly after slaughter, and were transported in a PBS buffer (pH 7.4). The nasal cavity was exposed with a sagittal cut along the nasal septum (to remove the mucosa from the nasal cavity). This was performed with curved surgical scissors and a Molt detacher, as described previously [23,24].

##### In Vitro Evaluation of Mucoadhesive Strength

The mucoadhesive strength was measured by detachment of the nasal mucosa from the formulations (TTC_HG4%_ and TTC_NLC-HG4%_), using a TA-XT Plus texture analyzer (Stable Micro Systems, UK) operated in tension mode. Firstly, the mucosal tissue was fixed in the tissue support of the A/MUC probe (Figure 1A) and moistened with artificial mucin [25] for 5 min. The samples (TTC_HG4%_ and TTC_NLC-HG4%_) were placed on the mucoadhesion probe (A/MUC) (Figure 1B).

The test was started with a constant tension speed of 0.1 mm·s^−1^, until the formulation touched the mucosal surface, followed by maintaining a compression force of 0.5 N for 20 s, at 32 ± 0.5 °C. A traction speed of 0.1 mm·s^−1^ was then applied, until the formulation detached from the mucosa [22]. The mucoadhesion work (calculated from the area under the force vs. distance curve) and the detachment force (the force required to detach the mucosa formulation) were computed using Exponent software (Stable Micro Systems, UK). Five replicates were measured.

##### In Vitro Permeation Studies

In vitro permeation assays were performed using a Franz-type vertical diffusion system (Phoenix™ DB-6 Manual Sampling System, Hanson Research, Los Angeles, CA, USA) with a permeation area of 1.77 cm^2^ and 16 mL acceptor compartment, at 32.5 °C [21], filled with 5 mM PBS (pH 7.4) plus 5% Tween 80. This provided sink conditions throughout the experiment, ensuring that the drug remained solubilized. The nasal mucosa thus prepared were visually inspected, and samples presenting any damage were discarded. The mucosal barrier was placed between the donor and acceptor compartments, followed by the addition of the buffer solution (1 mL) to the donor compartment.

After tissue selection, the buffer solution in the donor compartment was replaced by 300 mg of TTC_HG4%_ or TTC_NLC-HG4%_, and magnetic stirring was applied at 300 rpm. At predetermined intervals, aliquots of the receptor solution (300 μL) were removed and analyzed by HPLC as described. The volumes removed from the acceptor compartment were immediately replaced with buffer solution. The experiments were carried out for 5 h, with five replicates.

The cumulative amount of TTC permeated across the porcine nasal epithelium was plotted as a function of time, with analysis according to passive diffusion, as performed in another permeation study of nasal mucosa [26]. The steady-state flux (*J_ss_*) was calculated according to Fick’s first law (Equation (2)). This mathematical model is well known for the analysis of permeation assays, enabling comparison of different studies [27].
(2)$$J_{ss}=\frac{\Delta Qt}{\left(\Delta t\times A\right)}\;\left[\mu g/{\mathrm{cm}}^{2}/h\right]$$
where:

∆*Qt* = Difference of permeated amount of drug

∆*t* = Difference of measurement time points (h)

*A* = Permeation area (cm^2^)

#### 2.4.4. Cell Culture and In Vitro Cytotoxicity Assays

Human neuroblastoma cells (SH-SY5Y) were cultivated in DMEM/F12-Ham supplemented with 10% FBS, 1% antibiotic (penicillin and streptomycin), and 2 mM L-glutamine. The cells were grown in culture flasks (75 cm^2^) under a humidified atmosphere with 5% CO_2_, and at 37 °C. After reaching 80% confluence, the cells were detached by treatment with 0.25% trypsin and 0.5 mM EDTA, to repeat the cell culture or to perform the experiments.

The SH-SY5Y cells were transferred to 96-well tissue plates (2 × 10^4^ cells/well) and incubated for 24 h under a 5% CO_2_ atmosphere, at 37 °C. Viable cells were treated for 24 h using TTC_HG4%_, TTC_NLC-HG4%_, and free 4% TTC (TTC_free_). The formulations were diluted in a culture medium, in the concentration range from 4 × 10^−1^ to 4 × 10^−5^ mg/mL. The percentages of viable cells were determined by the MTT reduction assay [28]. Briefly, the wells were washed with pre-warmed PBS (pH 7.4), and a 200 µL volume of DMEM/F12-Ham with 0.5 mg/mL MTT sodium salt was added to each well. The plates were incubated for 4 h at 37 °C, followed by removal of the medium. Finally, a 150 μL volume of ethanol was added to each well to dissolve the formazan crystals, resulting in a purple solution. The fraction of viable cells was obtained by quantification of the original formazan content using a microplate reader (BioTek Instruments Inc., Winooski, VT, USA), at 570 nm, with the value converted to the percentage of viable cells. The IC_50_ was calculated using a dose–response curve with a variable slope, employing GraphPad Prism 5 software (GraphPad Software Inc., Northampton, MA, USA). The differences were considered significant at *p* < 0.05. In each experiment, the formulations were tested using nine wells for each concentration. The experiments were carried out at three different times [29].

#### 2.4.5. In Vivo Antinociceptive Assays

The protocol was approved by the UNICAMP Institutional Animal Care and Use Committee (#4004-1, 25^th^ September, 2015), following the instructions of the *Guide for the Care and Use of Laboratory Animals*. Male Swiss albino mice (30–35 g) were acquired from the Multidisciplinary Center of Biological Investigation of Laboratory Animals (CEMIB-UNICAMP). The animals were kept in cages and were acclimated for 7 days under 12/12 h light/dark cycles, with water and food ad libitum, at 22 ± 3 °C.

The tail-flick test was used to detect the antinociceptive effects of TTC_NLC-HG4%_, TTC_HG4%_, TTC_NLC-HG2%_, TTC_HG2%_, HG, and HG + NLC. For this, the tail of the animal (5 cm from the base) was exposed to heat (55 ± 1 °C) produced by the analgesimeter (Onda Científica, Campinas, Brazil). The latency time was recorded as the time (min) between the heating stimulus and the first movement to withdraw the tail, with comparison to the baseline (the time recorded before sample application). A maximum exposure cut-off time of 10 s was adopted, in order to avoid injury to the mice. About 0.1 g of each formulation was applied to the tail of the animal. After 30 min, the thermal stimulus was initiated and was repeated every 1 h [10]. The maximum possible effect (%MPE) was calculated according to Equation (3):(3)$$\mathrm{MPE}\;\left(\%\right)=\frac{\left(\;\mathrm{treshold}-\mathrm{baseline}\right)}{\left(\mathrm{cutoff}-\mathrm{baseline}\right)}\times 100$$

#### 2.4.6. Statistical Analysis

Statistical analyses were carried out using GraphPad Prism v. 6.01 (GraphPad Software, Inc., Northampton, MA, USA). The Student’s *t*-test (*p* < 0.05) was used to compare the mucoadhesion and permeation results for the hybrid systems and the PL407 gels. The amount of TTC permeated over time was submitted to linear regression analysis. The effects of the tetracaine formulations on cell viability were analyzed using two-way ANOVA and the Bonferroni post hoc test. Nonlinear regression analysis was used to determine the IC_50_ values in the cytotoxicity assays. One-way ANOVA followed by the Tukey post hoc multiple comparisons test (*p* < 0.05) was used to analyze the data (areas under the curves) from the tail-flick tests.

## 3. Results and Discussion

### 3.1. Characterization of TTC_NLC_ as the Lipidic Component of the Hybrid Hydrogel

The anesthetic selected was tetracaine (also known as amethocaine and commercially available as Ametop^®^) at a dose of 4%, based on previous clinical trials that demonstrated its anesthetic efficacy [30,31]. The TTC_NLC_ particles displayed submicron diameters, monodisperse size distribution, and good electrical charge repulsion, as shown in Table 1.

### 3.2. Preparation of Hybrid Hydrogels

TTC_NLC-HG4%_, TTC_HG4%_, TTC_NLC-HG2%_, TTC_HG2%_, HG, and HG + NLC were successfully prepared and showed homogeneous aspects and pale white colors. All the formulations were stored at 4 °C, prior to subsequent analyses. TTC_NLC-HG2%_ and TTC_HG2%_ were only used in the in vivo study in order to determine whether lower anesthetic concentrations could provide efficacy similar to TTC_NLC-HG4%_ and TTC_HG4%_.

### 3.3. Rheological Analyses

Figure 2A,B shows the rheograms obtained for HG, HG + NLC, TTC_NLC-HG4%_, and TTC_HG4%_. All the formulations were found to be thermosensitive, since changes in the viscoelastic properties occurred with an increase in temperature, with the materials becoming gels at the nasal mucosa temperature (32 °C) [32]. The sol-gel transition temperatures (T_sol-gel_) for HG and HG + NLC were 22.2 and 25.7 °C, respectively, showing that NLC incorporation increased T_sol-gel_ by 3 °C. Another important point, from comparison of the rheological profiles for TTC_NLC-HG4%_ and TTC_HG4%_, was the more pronounced shift of T_sol-gel_, highlighting the influence of incorporation of both TTC and NLC into the HG structural organization, in agreement with a previous report. Since the HG-based thermogelation process is driven by the self-aggregation of micelles, which requires the establishment of interactions between PEO and PPO units, the incorporation of TTC into the hydrogel increased the gelation point by 7 °C, possibly because the charged (free) TTC interacted with the surfactant molecules, disturbing the thermogelling process [12]. Similar effects were observed after the incorporation of the NLC into the HG, which indicated that both TTC and NLC influenced hydrogel formation and the maintenance of its structure, as described previously for other nanocarrier systems, such as chitosan-tripolyphosphate nanoparticles [33] and poly-ε-caprolactone nanoparticles [34] in association with PL407 hydrogels. This behavior was interesting, because the gelation process of TTC_NLC-HG4%_ occurred at a higher temperature (22.4 °C) than that of TTC_HG_, favoring its gelation when in contact with the nasal mucosa [35,36].

Frequency sweep analysis revealed viscoelastic behavior, with the predominance of G’ over G”, at 32.5 °C, for all the formulations, as shown in Figure 2C,D. The storage modulus (G’) of a viscoelastic material reflects its ability to store incoming mechanical energy [37]. Accordingly, the NLC had no pronounced effect on the response of the gel to the stress. On the other hand, the incorporation of TTC-loaded NLC into the hydrogels enhanced G’/G” by around 10–25 times, indicating that drug encapsulation reduced the disturbing effects caused by the incorporation of TTC into PL407 HG. An important point concerned the possible organization of the system, since the NLC were themselves coated with PL188 (a more hydrophilic analogue of PL407), allowing interactions between PEO units from PL188 and PL407, stabilizing the system and maintaining the structural organization of the hydrogel, even at high frequency values. Moreover, the predominance of G’ over G” showed the potential of TTC_NLC-HG4%_ to be topically administered at the nasal mucosa and, possibly, to resist the shear stress caused by mucociliary activity, resulting in a longer residence time and enhanced drug absorption [5,6].

### 3.4. Texture Profile Analysis (TPA)

TPA is an essential tool to investigate the effects of different compositions of formulations on their mechanical properties [38,39]. This analysis was performed to identify whether the TTC-loaded NLC could alter the structure of TTC_HG4%_.

As shown in Table 2, there was no difference (*p* > 0.05) between TTC_NLC-HG4%_ and TTC_HG4%_ in terms of the cohesiveness and elasticity parameters. On the other hand, TTC_NLC-HG4%_ was significantly (*p* < 0.01) more adhesive (1.062 ± 0.056 N) than TTC_HG4%_ (0.534 ± 0.154 N).

Elasticity is a mechanical property related to the reversible deformation of a formulation submitted to compression stress, following a wait time between compressions, while cohesiveness indicates how well the formulation tolerates a second deformation, relative to its resistance under the first deformation [37,40]. It could be seen that the incorporation of TTC-loaded NLC into the PL407 hydrogel did not affect the ability of the gel to stretch and reorganize its structure, nor did it affect the restructuring after a subsequently applied force. These findings reinforced the hypothesis that the NLC were distributed between the poloxamer micelles, without affecting the packaging and entanglement of the micellar structure [12].

In the case of adhesiveness, the value can be affected by the hydrogel viscosity. This could explain the higher adhesiveness of TTC_NLC-HG_, since the value of the phase angle (tan δ = G”/G’) of TTC_HG_ was slightly higher than that of TTC_NLC-HG_. When tan δ becomes higher, the elasticity (solid character) of the formulation decreases, while the viscous behavior (liquid character) increases [41]. In addition, higher adhesiveness is indicative of a longer contact time between TTC_NLC-HG_ and the nasal mucosa, leading to better clinical performance [42,43].

### 3.5. In Vitro Evaluation of Mucoadhesive Strength

In the pharmacotechnical field, mucoadhesion can be defined as the interaction between a formulation and the mucin layer. This interaction is firstly achieved by contact, spreading, and swelling of the formulation on the mucosa, followed by consolidation involving chemical or physical bonds between the molecules of the formulation and the mucin [44,45]. In this way, a mucoadhesive drug delivery system is expected to adhere and release the drug to a specific mucosa during an extended period [46].

Texture analyzers are commonly used to measure the detachment of hydrogels from artificial mucin [22,37,40,44,47]. This method allows evaluation of the different stages of the mucoadhesion mechanism, such as the contact and the strength of interaction between the formulation and the mucin, very closely mimicking real mucoadhesion conditions [48].

The most commonly evaluated mucoadhesion parameters are (i) the maximum force necessary to separate the probe from the tissue (maximum detachment force), and (ii) the total amount of force involved in removing the probe from the tissue (mucoadhesion work) [38,49]. Therefore, these two parameters were used to compare the mucoadhesion parameters of TTC_NLC-HG4%_ and TTC_HG4%_, as shown in Table 3.

The incorporation of TTC-loaded NLC into HG did not change the force required to detach it from the nasal mucosa. The presence of NLC in the hybrid hydrogel could not interfere with the flexibility of the polymer chain, due to its high density of crosslinks. Polymer chains with high flexibility can provide greater interfacial contact and a more favorable environment for entanglement between the polymer and the mucin molecules [13,37,47], favoring the mucoadhesion process. These data corroborated the TPA and rheology results, suggesting that the aggregative properties of PL407 were not affected in TTC_NLC-HG4%_.

Interestingly, the mucoadhesion work of TTC_HG4%_ was higher than that of the hybrid system, with TTC_HG4%_ taking longer to complete its detachment from the mucosa during the probe withdrawal step. Since the detachment force was similar for the two formulations, the results suggested that the high mucoadhesion work of TTC_HG4%_ could have been related to the low internal resistance of its molecules for withstanding external forces without breaking. Thus, the intermolecular bond strength was weaker for TTC_HG4%_ compared to the TTC_NLC-HG4%_ formulation, resulting in the better mucoadhesive performance of the latter [40,50].

Carvalho et al. [6] reported that a formulation developed for topical nasal administration of an antiretroviral therapy in a liquid crystal precursor showed similar mucoadhesive performance in porcine nasal mucosa to that presented by TTC_NLC-HG4%_. Interestingly, the study also revealed that this formulation showed faster in vivo absorption of zidovudine when the drug was administered topically through the nasal mucosa compared to intravenous administration. This was suggested to be due to the anatomical, physiological, and histological characteristics of the nasal cavity, which provided mucoadhesion and, consequently, rapid absorption and onset of action [6]. An implication is that the present formulation could present fast and efficient absorption in vivo, leading to good clinical efficacy.

### 3.6. In Vitro Permeation Studies

Figure 3 shows the profiles for permeation of TTC_HG4%_ and TTC_NLC-HG4%_ across the porcine nasal mucosa, while Table 4 gives the steady-state flow (*J_ss_*) and lag time values calculated from the curves.

There was no statistical difference between the formulations in terms of the steady-state flux and lag time parameters (*p* > 0.05). However, the linear regression analysis showed that permeation across the nasal epithelium mucosa was significantly (*p* < 0.0001) higher for TTC_HG4%_, compared to TTC_NLC-HG4%_. Hence, the encapsulation of TTC in NLC did not influence the initial permeation of TTC, but the permeation could be sustained for longer, and with a lower flux. Similar results were reported previously for the fluxes of local anesthetics encapsulated in polymeric nanocapsules, which were also significantly lower than obtained for the same anesthetics incorporated in a commercial formulation [22].

Previous studies strongly suggest that an anesthetic reservoir is formed by the NLC lipids (resembling the lipids found in biological membranes), decreasing the number of free anesthetic molecules available to cross the epithelial barrier, while at the same time prolonging the anesthetic effect [51,52].

### 3.7. Cell Viability Assays

Despite the limitations of monolayer in vitro cell culture models, which can hinder correlation to in vivo studies, these are well-defined methods for evaluation of the potential cytotoxicity of drugs [53]. The SH-SY5Y cell line has been used in various pharmacology studies, including investigation of the cytotoxicity of aminoester and aminoamide local anesthetics [54]. Neuronal toxicity is an important issue for local anesthetics. The SH-SY5Y cells can simulate some biological features of neurons, such as the expression of voltage-dependent sodium channels Na(v)1.2 and Na(v)1.7. Therefore, these cells are frequently used in in vitro experiments to assess the cytotoxicity of local anesthetic agents [54,55,56,57]. The changes in SH-SY5Y cell viability after exposure to TTC (free, TTC_HG4%_, and TTC_NLC-HG4%_) for 24 h are shown in Figure 4.

The results only revealed a statistical difference (*p* < 0.05) for TTC_free_ at 4 × 10^−5^ mg, compared to the other formulations (TTC_HG4%_ and TTC_NLC-HG4%_). Nonlinear regression analysis was used to calculate the IC_50_ values for the TTC formulations (*p* > 0.05), which were 0.006, 0.008, and 0.01 mg·mL^−1^ for TTC_NLC-HG4%_, TTC_HG4%_, and TTC_free_, respectively.

The hybrid hydrogel significantly increased the cytotoxic effect of tetracaine towards the SH-SY5Y cells. As reported previously, encapsulation of TTC in NLC resulted in the TTC permeation flux being maintained for longer, which could have increased the bioavailability of TTC, consequently increasing the intrinsic toxicity of the drug.

Ribeiro et al. [58] investigated the cytotoxicity, after 24 h, of NLC loaded with lidocaine and prilocaine and incorporated in pectin-based films. It was found that the NLC did not influence the cytotoxicity of the films, but decreased the cytotoxicity of the drugs, suggesting that they had to overcome a double barrier in order to be released. However, the study was performed using fibroblasts (3T3) and keratinocytes (HaCaT) as cell models, which might explain the differences, compared to the present study, where the formulations were tested using neuroblastoma cells.

Muniz et al. [22] investigated the cytotoxicity of lidocaine and prilocaine (2.5%) encapsulated in nanocapsules incorporated in carbopol hydrogels. It was shown that hydrogels with encapsulated anesthetics were less cytotoxic than hydrogels with non-encapsulated anesthetics, suggesting a cytoprotective action of the nanocapsules. Nevertheless, the treatment was performed for only 3 h, while a period of 24 h was used in the present study, and it is known that in cell culture tests, loss of cell viability is directly proportional to the duration of exposure and the dose of the drug [59]. Furthermore, the longer exposure time allowed an increased release of the LA from the carriers.

### 3.8. Antinociceptive Tests

As shown in Figure 5A,B, TTC_NLC-HG4%_ demonstrated a longer duration of analgesia (34 h) compared to TTC_NLC-HG2%_ (26 h), TTC_HG4%_ (10 h), and TTC_HG2%_ (8 h). As expected, the formulations without TTC (HG and HG+NLC) presented no analgesia effect (Figure 5C). 

Both the concentration of TTC and its encapsulation in NLC influenced the duration of analgesia. As discussed above, the lipid composition of the NLC may have favored storage of TTC at the site of administration, consequently prolonging the in vitro permeation across the nasal mucosa, as well as the anesthetic effect in the mice [51,52].

Another study that evaluated the permeation of benzocaine and lidocaine from a hybrid hydrogel (NLC incorporated in glycerol–xanthan gum hydrogel) across the skin found slower permeation, while the anesthetic effect in mice was prolonged [52].

It should be noted that this antinociceptive test was performed with the application of the formulations on the skin of the tails of the animals. Many studies have observed that drugs permeate more easily across mucosas (nasal, buccal, sublingual, and gingival), compared to skin [60], due to the presence of the stratum corneum (SC) in the latter. The SC is a lipid matrix with a thickness of around 15 μm, composed of ceramides, cholesterol, and other free fatty acids, which provides the skin with a protective barrier that limits the penetration of particles into the organism [61].

The nasal mucosa is composed of squamous keratinized epithelium with a single layer of pseudostratified ciliated columnar respiratory cells, goblet cells, and the submucosal glands [62]. The low thickness of the nasal mucosa contributes significantly to faster nasal permeation of drugs, indicating the possibility of a better anesthetic effect of TTC_NLC-HG4%_ administered by the intranasal route.

In addition, the moisture present in the mucus-covered nasal surface favors the process of mucoadhesion of TTC_NLC-HG4%_, compared to the drier skin surface. Hence, the anesthetic effect of TTC_NLC-HG4%_ is enhanced by use of the intranasal route, since it is known that mucoadhesion can favor the rapid onset action of drugs [6].

Therefore, the findings of this work open up perspectives for the excellent anesthetic performance of TTC_NLC-HG4%_ in humans, given that previous studies have observed that the flux of an anesthetic can be used to predict the efficacy of an anesthetic in the preclinical phase [63,64].

For the formulations without TTC (PL407 hydrogel (HG) and NLC incorporated in HG (HG + NLC)), the thermal sensitivity of the animals changed abruptly in the first 60 min of the tail-flick test, as shown in Figure 5C. As expected, after one hour, no anesthetic effect was observed for the HG or HG + NLC samples. This behavior was probably due to the thermoreversible property of PL407 hydrogel [65,66], which undergoes polymerization at the body temperature, as confirmed by visible formation of a thin film at the site of application on the mouse tail. This film could have protected the tail from the initial heating, explaining the different response (higher MPE) at 30 min, but not at longer times, since local sensitivity was subsequently restored.

Overall, the results obtained here were very encouraging, showing that TTC_NLC-HG_ is a genuine candidate as a drug delivery formulation for use via the nasal route. Although an intranasal spray composed of 3% tetracaine plus 0.05% oxymetazoline already exists, its indication is limited to single maxilla restorative procedures in patients weighing at least 40 kg. In this case, the failure of TTC to anesthetize the posterior superior alveolar nerve was probably due to its inability to penetrate the more posterior regions of the maxillary sinus, compromising anesthesia in the region of the second premolar, with complete failure in the region of the first molar [3]. In the present work, the results showed that TTC_NLC-HG4%_ can prolong both TTC permeation and the duration of anesthesia, which should encourage its clinical evaluation after topical nasal administration, aiming at providing effective analgesia that allows more invasive dental procedures, such as multiple tooth restorations, simple extractions, and endodontic procedures.

From a different perspective, the drug delivery system designed in this work could also be used as a nose–brain delivery system for the treatment of diseases such as migraine, cluster headache, and trigeminal neuralgia, for which the nasal route is preferable [67,68,69,70,71,72].

## 4. Conclusions

TTC-loaded NLC were successfully incorporated in poloxamer PL407 hydrogel, with the formulation presenting desirable characteristics for the provision of topical, intranasal, needle-free anesthesia. The 4% TTC_NLC-HG_ hybrid formulation showed improved mechanical, rheological, and mucoadhesive properties, together with low cytotoxicity, sustained in vitro permeation, and extended in vivo anesthesia, compared to tetracaine directly loaded in the hydrogel. These findings support its clinical evaluation in order to establish whether it is able to provide effective, topical, intranasal, needle-free anesthesia in dentistry.

## Figures and Tables

**Figure 1 pharmaceutics-13-01760-f001:**
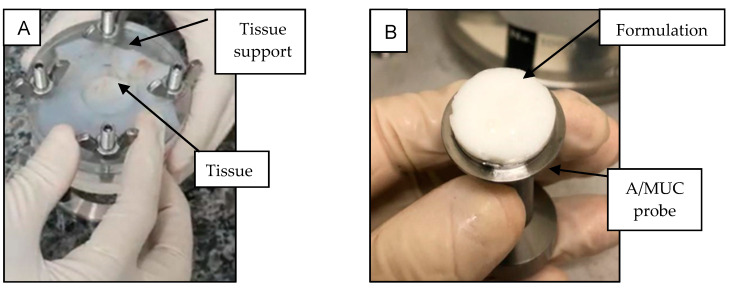
Preparation of the porcine nasal mucosa for the mucoadhesion test: (**A**) biological tissue fixed on the tissue support; (**B**) formulation fixed on the A/MUC probe.

**Figure 2 pharmaceutics-13-01760-f002:**
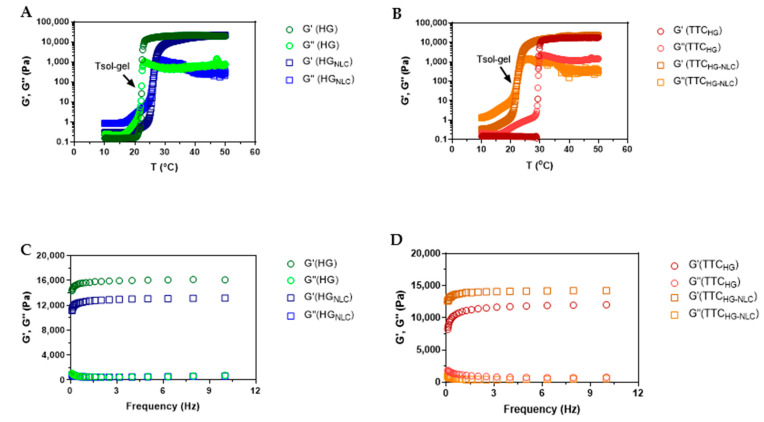
Rheology measurements. Storage modulus (G’) and loss modulus (G”): as a function of temperature for (**A**) HG and HG + NLC and (**B**) TTC_NLC-HG4%_ and TTC_HG4%_; and as a function of frequency for (**C**) HG and HG + NLC and (**D**) TTC_NLC-HG4%_ and TTC_HG4_, measured at 32.5 °C.

**Figure 3 pharmaceutics-13-01760-f003:**
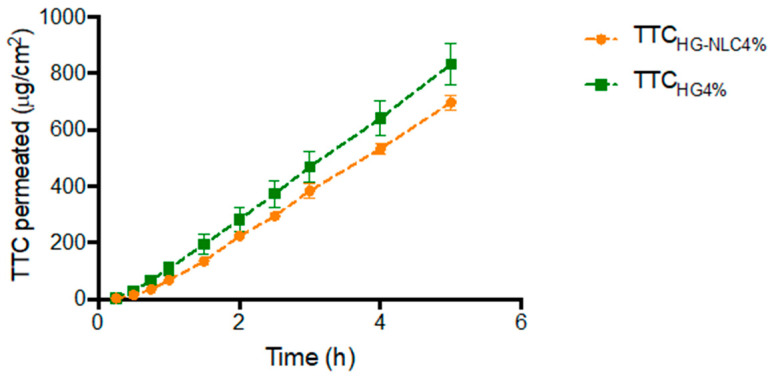
Profiles for permeation of the tetracaine-loaded formulations (TTC_NLC-HG4%_ and TTC_HG4%_) across the porcine nasal mucosa (mean ± SE, *n* = 5). Linear regression analysis between curves: *p* < 0.0001.

**Figure 4 pharmaceutics-13-01760-f004:**
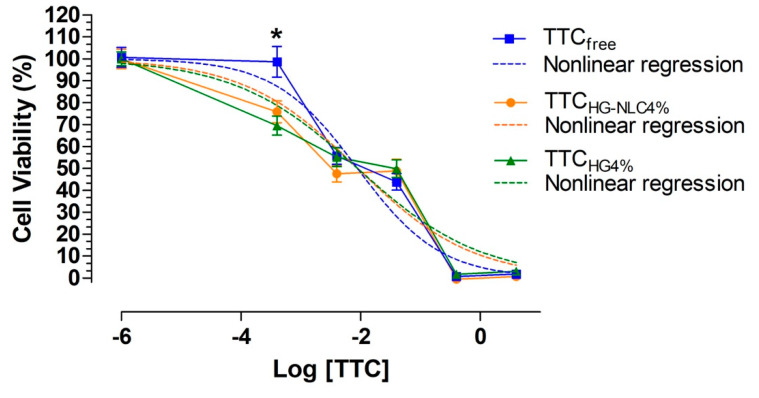
Effects of 4% TTC (TTC_free_), TTC_HG4%_, and TTC_NLC-HG4%_ on the viability of SH-SY5Y neuroblastoma cells (mean ± SEM, *n* = 3). *Y*-axis: cell viability in %, relative to the control group. *X*-axis: log of the drug concentration (%). The dotted lines show the nonlinear regression curves (log(inhibitor) vs. normalized response, variable slope). * Statistical differences between TTC_HG_ and the other TTC formulations (using two-way ANOVA and the Bonferroni post-hoc test). In each experiment, the formulations were tested using nine wells for each concentration, at three different times.

**Figure 5 pharmaceutics-13-01760-f005:**
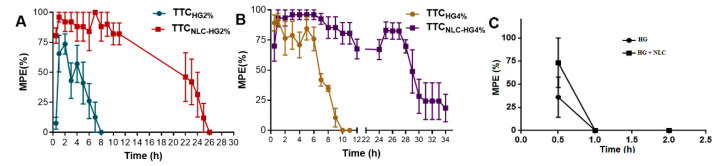
Anesthetic effect (tail-flick test) in mice treated with: (**A**) TTC_HG2%_ and TTC_NLC-HG2%_; (**B**) TTC_HG4%_ and TTC_NLC-HG4%_; and (**C**) PL407 hydrogel (HG) and NLC incorporated in HG (HG + NLC). The results are expressed as maximum possible effect (%MPE) vs. time (h). The data are shown as mean ± SEM (*n* = 6).

**Table 1 pharmaceutics-13-01760-t001:** Particle size, polydispersity index (PDI), zeta potential (ZP), and encapsulation efficiency (%EE) of TTC_NLC_. The values are shown as mean ± standard deviation.

Formulation	Size (nm)	PDI	ZP (mV)	%EE
TTC_NLC_	222.2 ± 2.6	0.154 ± 0.020	−30.1 ± 0.3	63.7 ± 4.2

**Table 2 pharmaceutics-13-01760-t002:** Mechanical properties of TTC_HG4%_ and TTC_NLC-HG4%_, at 32 °C, determined by texture profile analysis. The values are shown as mean ± standard deviation.

Formulation	Cohesiveness (Dimensionless)	Adhesiveness (N)	Elasticity (Dimensionless)
TTC_HG4%_	0.920 ± 0.062	0.534 ± 0.154 *	0.987 ± 0.009
TTC_NLC-HG4%_	0.916 ± 0.016	1.062 ± 0.056	0.989 ± 0.006

* *p* < 0.05 (unpaired *t*-test). Each parameter was analyzed separately (*n* = 5).

**Table 3 pharmaceutics-13-01760-t003:** Mucoadhesion parameters of the formulations.

Formulation	Mucoadhesion Parameters	
	Detachment Force (N)	Mucoadhesion Work (N·mm)
TTC_NLC-HG4%_	0.108 ± 0.045	0.051 ± 0.007
TTC_HG4%_	0.128 ± 0.029	0.071 ± 0.015 *

* *p* < 0.05 (unpaired *t*-test). Each parameter was analyzed separately (*n* = 5).

**Table 4 pharmaceutics-13-01760-t004:** Mean (±SD) values of the steady-state flux (*J_ss_*) and lag time for permeation of tetracaine from TTC_HG_ and TTC_NLC-HG_ across the nasal mucosa.

LA	Formulation	*J_ss_* (µg·cm^−2^·h^−1^)	Lag Time (h)	R^2^
TTC	TTC_NLC-HG4%_	150.23 ± 9.76	0.463 ± 0.120	0.998 ± 0.00135
	TTC_HG4%_	176.42 ± 29.49	0.370 ± 0.154	0.995 ± 0.0073

Each parameter was analyzed separately (*n* = 5). R^2^: coefficient of determination of the linear regression model.

## Data Availability

The data presented in this study are available on request from the corresponding author.

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
