# Peer review of "Mucoadhesive, Thermoreversible Hydrogel, Containing Tetracaine-Loaded Nanostructured Lipid Carriers for Topical, Intranasal Needle-Free Anesthesia"

_pharmaceutics, 2021, doi:10.3390/pharmaceutics13111760_

Round 1

Reviewer 1 Report

This study reports the development and characterization of tetracaine-loaded lipid nanocarriers incorporated into a thermoreversible poloxamer hydrogel for topical application. This hybrid system is characterized to determine its rheological characteristics, mucoadhesiveness, drug permeability across nasal mucosa, cytotoxicity in human neuroblastoma cells, and ability to provide analgesia. Overall, I like the study. The report is generally well written and shows the potential impact of this system in providing better analgesia through attachment to mucosa (nasal mucosa is targeted here) and enhanced prolongation of anesthetic effect. Comments provided below are mostly meant to help improve the clarity of the paper, but a few key issues regarding the discussion and conclusions must be addressed.

  1. Lines 65-66: Not sure what the “monodisperse size distribution” numbers are showing. What is 0.154? This needs to be clarified.
  2. Lines 123-124: Significantly more information is needed in regards to the rheology method. What was the plate size, gap, and cone angle? Over what time period was the temperature sweep conducted? Was there a hold between temperature changes? How was the transition temperature determined from the temperature sweeps (there are many different methods used for this, so it should be specified). Why was 32.5°C chosen for the frequency sweeps? The authors should be clear about what state the formulation was in at this temperature and why 32.5°C is an important temperature to consider.
  3. Lines 150-162: How do you know that this technique specifically measured the detachment of the hydrogel from the artificial mucin? It seems quite possible that it could measure either the detachment of the hydrogel from the probe surface or the mucin on hydrogel detachment from the underlying nasal mucosa. How do you know for sure what the measurements determine?
  4. Line 193: It is not clear why human neuroblastoma cells are relevant in this study. Why this particular cell line?
  5. Line 211: The n values noted here are not clear. It sounds like 3 independent experiments were conducted and thus n=3. Individual wells do not typically count as an n in studies with cell lines. It would be different if these were primary cells from different donors. See https://pubmed.ncbi.nlm.nih.gov/29617358/ for more guidance on this.
  6. Lines 277-286: I simply do not see that there is any major difference in G’ and G” between the four formulations. If NLC is not different from HG, how is TTC different? They are all within the same order of magnitude. Seems like the authors are trying to read more into the data than is there.
  7. Line 410: “It is possible to observe a slight reduction in the cytotoxicity of TTCHG4%...” Looking at figure 4, the graphs for HG4% and HG-NLC4% look exactly the same. Again, it seems the authors are reading more into their data than is being presented. Also, I don’t understand the nonlinear regression curves for these two formulations. They don’t match the data, so making any inferences based on them is pure speculation.
  8. Line 490: Suggesting that this formulation is “biocompatible” because of low cytotoxicity is a stretch. There is more to being biocompatible than just not killing the cells. This should be removed.
  9. Lines 491-492: The last statement of the conclusions is too far of a reach. Unless it is proven in the paper, this data does not suggest that the formulation could be successful for other uses, and it certainly doesn’t “endorse” its use.

Reviewer 2 Report

The research article entitled, “Mucoadhesive thermoreversible hydrogel containing tetracaine-loaded nanostructured lipid carriers for topical intranasal needle-free anesthesia”, endeavors to provide a safe, effective, painless, and prolonged analgesic treatment for dental procedures. The hydrogel NLCs formulation showed reduced cytotoxicity, and enhanced permeation capacity and anesthetic efficacy.

Although the authors have tried to include varied aspects on characterization of thermoreversible hydrogel containing tetracaine-loaded nanostructured lipid carriers, yet the article lacks on several vital fronts. A succinct account is as follows:

  1. The authors have not provided sufficient details on the preparation of tetracaine-loaded nanostructured lipid carriers. The reference paper (no. 11) provided by the authors for the preparation of tetracaine NLCs has not been published yet. The authors are advised to provide detailed procedure behind the selection of lipids and the quantities opted for formulating NLCs. Also, the graphs confirming the nanosize and zeta potential of the formulation should be added along with a tabular representation with standard deviation/average value of three independent measurements.
  2. The authors loaded the tetracaine-loaded nanostructured lipid carriers into thermoreversible hydrogels. On what basis the authors selected the dose of tetracaine, quantities of polymer employed and the formulation conditions like stirring time? The authors are advised to clarify the same.
  3. The authors used hybrid hydrogels containing 25 mL of 4% tetracaine hydrochloride-loaded hydrogel (TTCHG4%) plus 25 mL tetracaine-loaded nanostructured lipid hydrogel (TTCNLC4%). The authors need to clarify the reason behind using TTCHG4% in hybrid systems.
  4. Figures resolution should be improved. Some figure captions also need revision in terms of significance value. The x-axis of Figure 4 should be revised as it looks quite congested.
  5. Figure 2 is missing in the text as well in the manuscript. Authors please be careful.
  6. The authors are advised to use the abbreviation at the first instance in the manuscript. Also, the abbreviation used should be consistent throughout, like authors used P407 and PL407 for poloxamer P407.
  7. Stability studies of the optimized hydrogels at different conditions will be important to evaluate. The authors need to justify the same.
  8. The results and discussion section needs improvement. There are long sentences with commas that make reader confused and not follow the flow of the study.
  9. The authors need to revise the manuscript and need improvement in presentation for better clarity. The manuscript contains some syntactical and typographical errors, which need to be critically examined and corrected.
